# Investigating the influence of language teachers' constructivist self-efficacy on their practice of constructivism in Ghanaian language and culture instruction

Ernest Nyamekye [1‡]*, Seth Asare-Danso[1‡], Emmanuel Amo Ofori[2‡]

**1** Department of Arts Education, University of Cape Coast, Cape Coast, Ghana, **2** Department of Ghanaian Languages and Linguistics, University of Cape Coast, Cape Coast, Ghana

‡ EN, SAD, and EAM are joint senior authors.
* ernest.nyamekye@ucc.edu.gh

## Abstract

The education system in Ghana is undergoing a transition from a behaviorist instructional philosophy to a constructivist one, aiming to produce learners who can actively contribute to nation-building. Nonetheless, given the heavy demands on teachers regarding this abrupt shift into constructivist teaching, there is a need to examine teachers' sense of efficacy in relation to the enactment of the core principles of this novel instructional philosophy—i.e., social, cognitive, and critical constructivism— laid down in the newly introduced standards-based curriculum. An explanatory sequential mixed method was used to obtain data from basic school teachers in the Sunyani-West Municipal of Bono Region, Ghana. Using adapted teacher self-efficacy and constructivist learning environment scales, quantitative data were gathered from 104 teachers. Qualitative data were also gathered from 15 conveniently sampled language teachers to augment the quantitative findings. Using partial least squares structural equation modelling, a significant positive association was discovered between teachers' efficacy and the practice of social and cognitive constructivism. Nonetheless, teachers' efficacy did not statistically predict their practice of critical constructivism. The qualitative results showed that sociocultural concerns probably accounted for the insignificant association between efficacy and critical constructivism. It was therefore concluded that sociocultural norms designed for bringing up a child in Ghana tend to inhibit the enactment of critical constructivism. The study recommends that the National Council for Curriculum and Assessment, in partnership with the Ghana Tertiary Education Commission, should update teacher professional development programs in universities and colleges of education to incorporate constructivist principles, particularly critical pedagogy, aiming to produce competent teachers capable of fostering learners' autonomy, critical thinking, and problem-solving skills as outlined in the Standards-Based Curriculum (SBC).

**Data availability statement:** The data underlying the results presented in the study are available from https://doi.org/10.7910/DVN/BUKKEB

**Funding:** The author(s) received no specific funding for this work.

**Competing interests:** The authors have declared that no competing interests exist.

## 1. Introduction

The need for change in the philosophy of teaching—from traditional behaviourist teaching to constructivist instruction—has become an issue of concern for many African and low and middle-income countries over the past two decades. The global interest in the constructivist teaching philosophy emanates from emerging concerns about the limitations associated with the traditional behaviourist approach to teaching. This traditional teaching approach has been criticized for its premium emphasis on teacher-centeredness. The widely held belief about this instructional philosophy is that its core principle—that is, the principle of reinforcement—fails to adequately address the complexities of critical thinking, problem-solving, and decision-making. Consequently, numerous countries and concerned agencies have advocated for the need to shift from the behaviourist philosophy of education to the constructivist philosophy of teaching to improve the quality of education and, most importantly, remedy the educational limitations associated with the behaviourist instructional philosophy [1,2]. Several African countries such as South Africa [3], Namibia [4], Tanzania [5,6], Nigeria [7], Ghana [8,9], Eritrea [10], and some Asian countries like, Cambodia [11], etc., have, therefore, devised educational reforms in which the constructivist teaching approach has been adopted as the new educational approach capable of ensuring quality education.

Leveraging the academic benefits associated with the constructivist philosophy is, obviously, one of the core reasons why the Ministry of Education (MoE), in collaboration with the National Council for Curriculum and Assessment (NaCCA) of Ghana, introduced the SBC in the year 2019. As clearly stated in the Ghanaian language and culture curriculum (GHLCC), teaching and learning are to be grounded in the constructivist philosophy. The underlying aim of this philosophical shift is to:

> …develop individuals who are literate, good problem solvers, have the ability to think creatively and have both the confidence and competence to participate fully in the Ghanaian society as responsible citizens, locally and globally [12]

While the above goals demonstrate a promising educational future for basic school children in Ghana, extensive literature indicates that the implementation of this instructional approach has been a challenge to teachers as compared to the traditional approach [1,13,14]. This echoes the position that the shift from the behaviourist teaching philosophy to constructivist teaching "lays a significant burden on teachers to adapt and modify their teaching" [14]. Consequently, it has been shown that though various educational reforms have advocated for the adoption of the constructivist approach, there is resistance to change among teachers because they constantly adopt teacher-centred teaching approaches as a result of the perceived complexities and challenges associated with the constructivist approach [15–19]. Scholars such as Nyamekye, Zengulaaru [20] have argued the many basic school teacher lack the requisite knowledge regarding the core principles of constructivist instruction and, as a result, their instructional practices are usually at variance with what demands of the current NaCCA curriculum which advocates for the adoption of constructivism. Studies such as that of Ampadu and Danso [21] share the view that the rare observance of constructivist instructional principles among Ghanaian teachers is caused by certain cultural issues which align with the traditional teaching principles among some teachers in Ghana. Prominent among this culture is the culture of acknowledging correct answers only and the culture of individual work. These authors have shown that teachers have not fully embraced the essence of encouraging collaborative learning to encourage students to share and co-construct learning among themselves. From a sociocultural point of view, Nyamekye, Zengulaaru [22] and Nyamekye, Mutawakil [23] have argued that the sociocultural values of most Ghanaian cultures, which

subconsciously manifest themselves in teachers' instructional practices, do not support knowledge construction among students. Their argument is contingent on the fact that the sociocultural lens of Ghanaian teachers regarding the cognitive ability of children is quite primitive; they tend to perceive children as immature beings and for that matter, it is their responsibility as knowledgeable adults to always implant knowledge in their young learners. This sociocultural worldview has been viewed as a crucial impediment of the execution of constructivism in Ghana.

Several studies have also shown that contextual and school-related factors such as large classroom size, and instructional resources, among others have made the execution of constructivism an arduous task for teachers. A preliminary study conducted by Nyamekye, Zengulaaru [20] in the Sunyani West Municipal indicates that the practice of constructivism among basic school teachers is very rare given the difficulties created by the aforementioned contextual factors. Given the existing concerns regarding the implementation of constructivism, we deemed it necessary to explore this research problem further using Bandura's self-efficacy theory. Understanding the implementation of constructivism and its variants (cognitive, social, and critical constructivism) from the perspective of the self-efficacy theory was necessary because teachers' behavioural disposition is a necessary condition for their academic engagements. Their perceived self-efficacy – i.e., their judgement of how effectively they can accomplish a given educational task [24] – for instance significantly affects their attitudes, motivation, and, most importantly, their success in that particular task [25]. People's way of thinking and their emotional reactions to situations could be affected by their sense of efficacy. This implies that teachers with a higher level of self-efficacy are more likely to be relaxed and productive amid challenging issues. It follows therefore that those with a low sense of efficacy are always pessimistic about situations because they feel that the work they are supposed to carry out is more challenging than it may be. With these pessimistic thoughts in mind, they are more likely to fail in accomplishing a given task because these thoughts increase their stress and anxiety levels, and most importantly, narrow their confidence necessary for overcoming a problem [26]. Based on this theoretical assumption, we were compelled to assume that teachers' sense of efficacy could have a compelling influence on their adoption of constructivist principles. Investigating the current research from this theoretical perspective provides a novel understanding of how behavioural issues on the part of teachers could further explain their(un)willingness to employ the constructivist instructional in the teaching and learning of the indigenous languages of Ghana. To achieve the aforementioned research objective and contribution, the study employs a statistical modelling approached (i.e., partial least squares structural equation modelling), complemented with interview data to explain how teachers' sense of efficacy could explain their actual execution of constructivism in language teaching.

## 2. Literature review

### 2.1. Constructivism as an instructional philosophy

Constructivism, a learning theory, is commonly approached from a cognitive standpoint, often seen as a response to the information processing theory that explores how the human brain encodes information for storage [27]. Similar to other cognitive learning theories, it is a conceptual framework for understanding how prior information in human memory becomes the foundation for acquiring new knowledge [28,29]. It challenges the traditional learning environment, rejecting the linear transmission of knowledge from teacher to students. Advocates of constructivism argue that the transfer of knowledge from an expert to learners is not the most effective way to acquire knowledge; rather, learners benefit more from actively participating in constructing their own knowledge. Constructivism promotes a learner-centered

approach, with the teacher playing the role of a facilitator or educational guide [30–33]. While many cognitive theories stress active learning, constructivism places particular emphasis on learners actively constructing their understanding of educational topics.

Constructivism, as a theory of learning and instruction, manifests in various forms such as cognitive, social, and critical constructivism. Cognitive constructivism, also known as individual or psychological constructivism, focuses on how individuals create knowledge through existing schemas [27]. Piaget [34] credited with the cognitive constructivist perspective, argues that knowledge is not merely received and absorbed but is constructed through coordinated cognitive activities based on experiences with the surrounding world. Individuals, according to Piaget, develop mental models or schemas through their interactions with the environment. Another variant of constructivism is social constructivism, introduced by Lev Vygotsky. In this theory, knowledge construction is viewed as a social process where individual efforts are complemented by significant contributions from others [35,36]. Unlike cognitive constructivism, social constructivism emphasizes the collaborative aspect of learning, where the interactions with others play a vital role in the construction of knowledge.

Critical constructivism, as a variant of constructivism, follows a similar learning approach. Notably, the distinguishing aspect of critical constructivism is its strong focus on fostering 'critical thinking [37]. Educators within the critical constructivism framework prioritize equipping learners with 'emancipatory potentials [38]. Therefore, the emphasis lies on guiding students to develop critical thinking skills, enabling them to reflect on societal experiences and analyze social structures and powers. Critical constructivists advocate for change, aiming to contribute to the creation of a better society. Consequently, they promote an educational approach that goes beyond acquiring content knowledge to instill awareness of societal or national issues in learners. In a critical constructivist learning environment, students are often tasked with independent or collaborative reflection on social issues, providing constructive contributions toward resolving these challenges [37,39].

In the education system of Ghana, the basic school curriculum advocates for constructivism. As specifically outlined in the SBC for basic education in Ghana, teachers are to apply the tenets of constructivism in their instructional activities. With regards to cognitive constructivism, the curriculum adopts the cognitive constructivism as it acknowledges that "Children have in-built potentials to develop and acquire new language while approximating grammatical structures as they learn to speak" [40]. The education system also adopts social constructivism as it encourages teachers to "…promote interaction with each other and make learners active in constructing their own knowledge, thoughts and experiences". Finally, critical constructivism also has a place in the curriculum because it encourages teachers to adopt innovative learning strategies capable of empowering learners to become autonomous learners. It is on this basis we intend to examine teachers' practice of critical, social, and cognitive constructivism.

## 2.2. Self-efficacy as a predictor of teachers' instructional practices

Teachers play a crucial role in implementing new curriculum strategies. Their perceived self-efficacy, which refers to their judgment of their ability to effectively complete an educational task [24], significantly impacts their attitudes, motivation, and, most importantly, their success in that specific task [25]. An individual's thought processes and emotional responses to situations may be influenced by their sense of efficacy. This suggests that individuals with higher self-efficacy levels are more likely to remain composed and productive when faced with challenges. Conversely, individuals with lower self-efficacy tend to approach situations with pessimism, perceiving tasks as more difficult than they actually are. These negative thoughts can hinder their success by increasing stress and anxiety levels and, crucially, by

diminishing the confidence required to overcome obstacles [26]. In this context, researchers in the field of education consistently show interest in and actively seek to assess the relationship between teacher efficacy and their conduct in the classroom, particularly in the context of new educational reforms.

Numerous studies highlight that teachers' sense of efficacy stands out as a key predictor of students' academic success, effective classroom management, and the effort invested in implementing innovative teaching strategies. According to Bandura [25], teachers' sense of efficacy, influencing the effort they dedicate to implementing educational innovations, stems from four primary sources: mastery experience (personal success), vicarious experience (inspiration from observing others), verbal persuasion (colleagues' words shaping decisions), and emotional states (how teachers' emotions affect their perceptions of capabilities). These sources collectively contribute to determining the level of teachers' self-efficacy, subsequently influencing their willingness and preparedness to adopt educational innovations.

Numerous studies have indicated that teachers' willingness to embrace change and implement new ideas as well as their enthusiasm for incorporating innovative teaching methods are closely linked to their self-efficacy [41–45]. This suggests that teachers with a high sense of efficacy have a greater likelihood of succeeding in implementing new educational or curriculum innovations compared to those with a low sense of efficacy [46–48] Consequently, the need for an investigation into teachers' efficacy concerning the implementation of educational reforms has become a significant concern for researchers in the field of education.

In recent literature exploring the integration of technology in language education, self-efficacy emerges as a prominent factor influencing various aspects of educators' engagement and acceptance of technology. Wang and Liao [49] conducted a study focusing on Chinese L2 students. Their study revealed that technological self-efficacy significantly predicts technology acceptance, explaining a substantial portion of its variance. Similarly, Wang and Pan [50] investigated Chinese English as a foreign language (EFL) instructors' self-efficacy level and its association with their classroom practices. The result of their study demonstrated that teacher self-efficacy notably influences work engagement, surpassing the impact of other factors like resilience.

A study by Zhi, Wang [51] also underscores the significant role of efficacy in technology adoption among Chinese EFL teachers. Through structural equation modeling and regression analysis, they found that self-efficacy significantly predicted a considerable portion of the variance in teachers' technology adoption rates. This emphasis on self-efficacy suggests its pivotal role in shaping educators' attitudes and behaviors towards incorporating technology into language instruction. Collectively, these studies highlight the crucial influence of self-efficacy in driving technological acceptance and adoption within language education contexts. Recognizing the importance of bolstering educators' self-efficacy can offer valuable insights for stakeholders, including teachers, trainers, policymakers, and officials, aiming to enhance technology integration efforts in language learning environments.

Concerning the influence of teachers' self-efficacy and the use of constructivist teaching methods, few studies exist. Temiz and Topcu [52] examined pre-service teachers' efficacy beliefs and their use of constructivist-based methods in their lessons. A positive correlation was found between teachers' sense of efficacy and their use of constructivist teaching methods. A follow-up interview data analysis confirmed the statistical revelation of this particular study. Overall, these scholars concluded that teachers with a high sense of efficacy tend to be more constructivist in their instructional practices compared to less efficacious. Using a statistical modelling approach Boz and Cetin-Dindar [53] conducted a related study that investigated, among other things, the influence of pre-service science teachers' self-efficacy beliefs on their implementation of constructivist learning environments. As with the findings of Temiz and

Topcu [52], this study also revealed that pre-service teachers self-efficacy has a significant positive influence on the implementation of constructivism.

The predictive influence of efficacy on constructivist-based teaching has also been confirmed in a large-scale study conducted in Singapore by Nie, Tan [13]. Using a structural equation modelling approach, these scholars particularly looked at the strength of influence of efficacy on both constructivist and didactic instruction. The results indicated that teachers' efficacy was more strongly related to constructivism than it was to didactic instructional approaches. Thus, these authors concluded that teachers with a stronger sense of efficacy are more likely to adopt constructivist teaching methods than those with a limited sense of efficacy.

Despite the existence of these studies, we have identified several gaps that warrant the conduct of this study, especially in Ghana. Geographically, there seem to be no existing studies looking at the influence of teachers' sense of efficacy and the use of constructivism methods in African contexts, especially in Ghana. Moreover, most existing studies did not look at the influence of efficacy on the variants of constructivist instruction. This study therefore explores the phenomenon a bit further, by specifically, investigating how teachers' sense of efficacy influences the various aspects of constructivism (i.e., cognitive constructivism, critical constructivism, and social constructivism). Most, importantly, a compelling gap that necessitates the conduct of the current study is that literature on teacher efficacy and the application of constructivism in language teaching is quite scarce in existing literature, especially in the African context. Given the identified geographical and knowledge gap, coupled with the issues of teachers' resistance to implementing the constructivist teaching philosophy across various educational contexts (it becomes crucial to comprehend the sense of efficacy among Ghanaian language (GHL) teachers regarding the adoption of constructivist teaching approaches specified in the recently introduced Standards-based Curriculum (SBC) at the elementary education level in Ghana.

## 2.3. Model conceptualisation and hypothesis development

The study was grounded in the self-efficacy theory of Bandura [25]. As previously discussed, the efficacy theory states that individuals' perceived confidence regarding the accomplishment of a particular task or the performance of a particular behaviour influences their actual practice. Thus, in the context of this study, we hypothesize that language teachers' practice of social, critical, and cognitive construction would, among other factors, be contingent on their perceived confidence regarding the execution of the aforementioned forms of constructivism. Fig 1 is a diagrammatic representation of the proposed model developed to ascertain the influence of efficacy on the practice of the three forms of constructivism.

Practically, this study seeks to achieve the above by statistically testing the following hypotheses:

- H1: Teachers' self-efficacy will influence their practice of cognitive constructivism in Ghanaian language teaching.

- H2: Teachers' self-efficacy will influence their practice of social constructivism in Ghanaian language teaching.

- H3: Teachers' self-efficacy will influence their practice of critical constructivism in Ghanaian language teaching

Based on the insights from the literature review, the research hypotheses for the current study seek to demonstrate that teachers' level of efficacy will have a predictive effect on

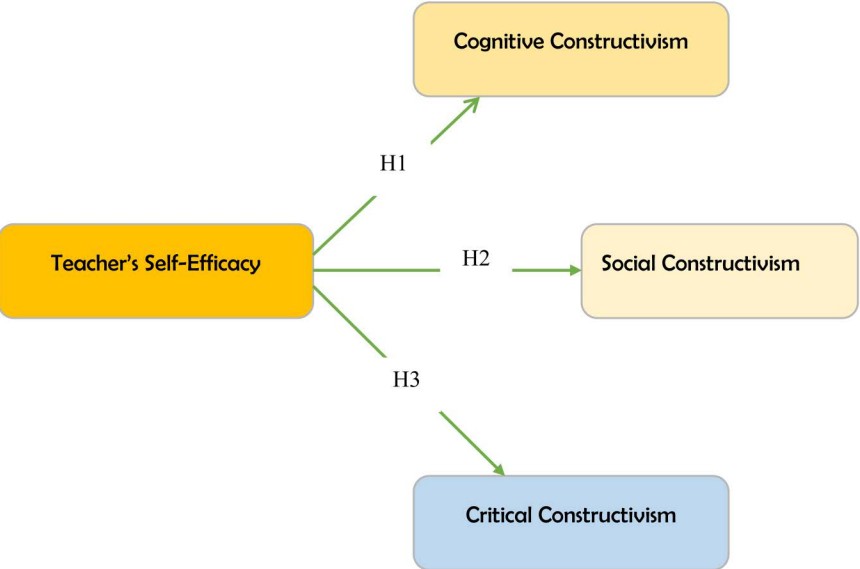

**Fig 1. Conceptual model.**

their practice of critical, cognitive, and social constructivism. In the context of this study, teachers with higher sense of efficacy are expected to demonstrate frequent implementation of the aforementioned constructivist practices, compared with low efficacious teachers

## 3. Methods

### 3.1. Study design, research area and selection of study participants

An explanatory sequential mixed method research design was adopted. As [54] rightly states, explanatory sequential mixed methods require an initial collection and analysis of quantitative data, followed by a collection of qualitative data to cross-validate the initial quantitative findings. The justification for using the explanatory sequential method for the current study is that after the collection and analysis of the quantitative results, the statistical results left unanswered questions that could only be answered through a qualitative inquiry. Based on this gap, we developed qualitative instrument for further data to explain the theoretical uncertainties revealed in the quantitative results.

The study was conducted in the Sunyani-West Municipal, located in the Bono Region of Ghana. The Sunyani-West Municipal is one of the 12 districts in the Bono region of Ghana. It has Odumase as its capital town. According to Nyamekye, Zengulaaru [20], the Sunyani-West municipality has 21 major settlements, including "Abesu, Abronye, Bofuor Adantia, Adei Boreso, Ahyiam, Ayakomaso, Chiraa, Chiraa-Asuakwa, Dumasua, Fiapre, Kantro, Kobedi, Kwabenakumakrom, Kwatire, Mantukwa, Nsesereso, Nsoatre, Odumase, Tainso, and Twumasi krom". The total number of basic schools located in the district is 61. Apart from Adantia, Chiraa, Dumasua, Fiapre, Kwatire, Nsoatre, and Odumase, most of the major settlements in the district are extremely remote.

For the collection of quantitative data, the researcher adopted the census survey technique to select all 104 participants. This technique for selecting research subjects is considered appropriate when the researcher intends to include all participants in a given population [55]. With regard to the qualitative data, a convenient sampling technique was used to sample participants for the study. This data collection procedure was adopted

because it allows the researcher to gather data from participants who are willing to participate in the research. Regarding the number of participants who were involved in the qualitative data collection, the researcher engaged as many as possible until data saturation was reached. Data saturation—a situation where themes recur in the information given by subsequent respondents [56]—usually occurs after interviewing 6 to 12 people [57]. In total, 15 language teachers willingly participated in the interviews.

## 3.2. Instrumentation

An adapted constructivist teacher self-efficacy scale (CTSE) of Uredi and Akbasli [58] was used to assess teachers' self-efficacy related to constructivist pedagogy. Samples of the questions are 'I can facilitate active learning and student exploration' and 'I can present content that are relevant to pupils' learning interests. The scale was measured on a five-point Likert scale ranging from strongly disagree (SD) to strongly agree (SA).

The constructivist learning environment survey (CLES) of Taylor et al. [59] was also adapted to measure the extent to which the three forms of constructivism, namely cognitive constructivism (CC), social constructivism (SC), and critical constructivism (CCT), were practised in the classroom. The CC items were adapted from the personal relevance scales (e.g., pupils are made to connect their previous knowledge of the language to the topic of the current lesson). The CCT scales were also adopted from the critical voice scales (e.g., I give pupils the authority to ask me to change my teaching methods). The SC scales were also adapted from the student negotiation scales (e.g., I encourage pupils to share their ideas with other students in the class). These scales were adapted from the CLES scale because the ideas in the personal relevance, student negotiation, and the critical voice constructs were developed from cognitive, social, and critical constructivism [59]. Since the original scale was design to measure the practice of constructivism in the science classroom, we replaced the phrase 'science classroom' with 'language classroom' where applicable in the adaptation process. A five-point Likert scale of frequency (ranging from never to always) was used to obtain teachers' responses regarding the practice of CC, SC, CCT. The original CLES is composed of five aspects of constructivism: personal relevance, shared control, critical voice, student negotiation, and uncertainty of knowledge. However, per the aims of the current study, we adapted questions from personal relevance, student negotiation, and critical voice because the aforementioned dimensions of the CLES align with cognitive, social, and critical constructivism, respectively.

For the qualitative data, a semi-structured interview guide was also used to gather further data on teachers' views on the practice of constructivism in language teaching. Questions related to teachers' practice of constructivism were focused, particularly, on the practice of cognitive constructivism, social constructivism, and critical constructivism. The researcher developed these qualitative constructivist-related interview questions by adapting questions from the Constructivist Environment Learning Survey (CLES) by Taylor, Fraser [59]. Concerning the practice of cognitive constructivism, interview questions related to how teachers enhanced knowledge construction by connecting learners' internalised linguistic skills to a particular topic were asked. Regarding the practice of social constructivism, the teachers were engaged in a discourse on how they enhance students' construction of knowledge through dialogism and knowledge sharing among their peers in the classroom. In relation to critical constructivism, the discourse was more focused on how teachers fostered autonomous learning by encouraging students to challenge power structures, including classroom norms that affect their learning, critical assumptions in books and other sources of knowledge, as well as teachers' instructional practices.

### 3.3. Validity and reliability checks

To ensure the reliability and validity of the research instruments to be adapted for this study, there was a pilot test in the Sunyani Municipality, where respondents had similar characteristics to those in the Sunyani-West district. A total of 150 questionnaires were pilot-tested to validate the research instrument. This number of questionnaires, according to Hertzog [60] is enough for the validation of an instrument for research. After pilot testing the instrument, Cronbach's alpha was used to assess the internal consistency of the initial version of the instrument. A good internal consistency value was obtained for TSE ($\alpha=0.766$; 5 items), CC ($\alpha=0.803$; 7 items), SC ($\alpha=0.791$; 5 items), and CCT ($\alpha=0.891$; 7 items). After the collection of the main data, PLS-SEM was used to evaluate the validity and reliability of the constructs. Particularly, the internal consistency, convergent, and discriminant validity values were used to confirm the suitability of the constructs and their respective items.

### 3.4. Data processing and analysis

The quantitative data gathered with a close-ended questionnaire were entered into the Statistical Product and Service Solutions (SPSS) version 25 for cleaning and analysis. After the data entry and cleaning in the SPSS software, the raw data were imported into the SMART PLS 4 software for statistical modelling analysis. Specifically, the statistical modelling begun with the measurement model assessment where the current study's model was subject to internal consistency, convergent validity, and multicollinearity assessment. Following the model quality evaluation, we assessed the structural model using the PLS-SEM bootstrapping method with 5000 subsamples. The analysis focused particularly on the path coefficients, the t values, the significant levels (p -values) as well as their corresponding effect sizes.

The follow-up interview data, recorded with Zoom Hn4 Pro 4-Track Portable recorder, were transcribed, coded, and analysed thematically with Audacity Transcription software and NVivo 11 software. The transcription of interview data was done selectively. In this regard, the researcher did not transcribe any paralinguistic features present in the record data. The selective transcription was deemed appropriate because the omission of paralinguistic features or non-verbal cues in the recorded data did not alter the meanings of the conversation [61,62]. Following the qualitative data analysis guidelines of Creswell, Hanson [54], we foremost read through to transcribed data meticulously. After familiarisng with the pertinent issues in the data, the researcher sorted out quotes that provide tangible explanations to the gaps in the quantitative results.

### 3.5. Ethical consideration

An ethical clearance was obtained from the University of Cape Coast's Institutional Review Board (ID-UCCIRB/CHAS/2023/97). After obtaining the ethical clearance, data collection commenced on 17th December 2023 and ended on 2nd April 2024. All schools were provided with an introductory letter that enabled the researcher to engage with the teaching staff as well as the learners. Also, the consent of all research participants was obtained before data collection. Teachers were made to sign a written consent form to confirm their participation in the study. Hence, under no circumstances were any of the participants forced to respond to the interviews or complete a questionnaire. All research participants were assured of the anonymity and confidentiality of the information they provided. The purpose of the research was communicated to the research participants; thus, they were informed that the information they provided would be used solely for academic purposes.

# 4. Presentation of results

## 4.1. Demographic background of respondents

Table 1 below presents the demographics of the respondents based on their age group, the highest academic qualification, and the number of years they have served as professional teachers.

A total of 104 teachers participated in the quantitative survey. Of these, 26.9% identified as male, while the remaining 73.1% identified as female, suggesting that the majority of the study's respondents were female. In terms of age distribution, 55.8% of the respondents were below the age of 30, 38.5% were within the age range of 30-34, and only 5.8% of the respondents were within the age range of 35-39. None of the respondents were in the age range of 40–44. The majority of the respondents (95.2%) had a diploma in basic education as their highest educational qualification. Only 4 respondents (3.8%) had a bachelor's degree. One teacher (1.0%) indicated a master's degree as his or her highest educational qualification. With regard to years of teaching experience, 87.5% have had one to five years of teaching experience, 9.6% have had six to ten years of teaching experience, and 1.9% have had 11 to 15 years of teaching experience. Only one teacher (1.0%) had taught for 16–20 years.

## 4.2. Measurement model

In PLS-SEM, assessment of the measurement model is a prerequisite for hypothesis testing [63]. It is done to ascertain the validity and reliability of the constructs in the proposed model. Particularly, internal consistency, convergent validity, and discriminant validity were assessed to ascertain the appropriateness of the model meant to test the study's proposed hypothesis. The subsequent subsections are devoted to presenting the statistical results for validity and reliability assessment.

**4.2.1. Internal consistency and convergent validity of constructs.** The specific statistical criteria needed for validity and reliability checks, according to Hair, Risher [63], are composite reliability, Cronbach's alpha ($\alpha$), and the average variance extracted (AVE). Table 2 and Fig 2 present the statistical results of the convergent validity of the constructs based on indicator loadings, Cronbach's alpha, composite reliability—i.e., the composite of Jöreskog [64]—and the AVE values.

**Table 1. Demographic characteristics of the research participants.**

| Variable | | Count | Percentage |
|---|---|---|---|
| | Subgroup | | |
| **Gender** | Male | 28 | 26.9% |
| | Female | 76 | 73.1% |
| **Age Range** | Below 30 | 58 | 55.8% |
| | 30-34 | 40 | 38.5% |
| | 35-39 | 6 | 5.8% |
| | 40-44 | 0 | 0.0% |
| **Highest Academic Qualification** | Diploma | 99 | 95.2% |
| | Degree | 4 | 3.8% |
| | Masters | 1 | 1.0% |
| **Years of Teaching Experience** | 1-5yrs | 91 | 87.5% |
| | 6-10yrs | 10 | 9.6% |
| | 11-15yrs | 2 | 1.9% |
| | 16-20yrs | 1 | 1.0% |

Table 2. Internal consistency and convergent validity of constructs.

| Construct | Indicator | Loadings | (α) | CR (rho_a) | (rho_c) | AVE |
|---|---|---|---|---|---|---|
| TSE | | | 0.795 | 0.829 | 0.858 | 0.548 |
| | tse1 | 0.739 | | | | |
| | tse2 | 0.833 | | | | |
| | tse3 | 0.634 | | | | |
| | tse4 | 0.750 | | | | |
| | tse5 | 0.732 | | | | |
| CC | | | 0.823 | 0.868 | 0.858 | 0.508 |
| | cc1 | 0.756 | | | | |
| | cc2 | 0.842 | | | | |
| | cc3 | 0.748 | | | | |
| | cc4 | 0.507 | | | | |
| | cc5 | 0.715 | | | | |
| | cc6 | 0.664 | | | | |
| | cc7 | 0.756 | | | | |
| SC | | | 0.760 | 0.792 | 0.836 | 0.508 |
| | sc1 | 0.667 | | | | |
| | sc2 | 0.802 | | | | |
| | sc3 | 0.806 | | | | |
| | sc4 | 0.685 | | | | |
| | sc5 | 0.577 | | | | |
| CCT | | | 0.845 | 0.819 | 0.861 | 0.557 |
| | cct1 | 0.829 | | | | |
| | cct2 | 0.813 | | | | |
| | cct3 | 0.696 | | | | |
| | cct4 | 0.793 | | | | |
| | cct5 | 0.570 | | | | |

The first consideration in assessing the measurement model is to check the appropriateness of the indicator loadings. Loadings that are equal to or above the recommended threshold of 0.70 are usually accepted because they are an indication that the underlying construct explains more than 50% of the indicator variable's variance [63]. Nonetheless, these authors further state that indicators that load as low as 0.40 could be retained in the model, especially when they do not affect other reliability values like those of the AVE. Thus, in the context of this study, as can be observed in Fig 2, it is safe to assume that the latent variables have met the recommended threshold for factor loadings.

The second consideration in the measurement model assessment is the internal consistency analysis. In this study, both Cronbach's alpha reliability and Jöreskog's [64] composite reliability were analysed. Composite reliability as an indicator of construct reliability is concerned with measuring the extent to which the underlying construct explains the variances in its indicators. The statistically accepted threshold for composite reliability is.70 [63]. As presented in Table 2, Cronbach's alpha values and the composite reliability values of TSE, CCT, CC, and SC are clear indications that the model's constructs have good internal consistency. It should also be noted that none of the composite reliability values is above 0.95. This indicates that there were no redundant items in each of the constructs. It further shows that undesirable response patterns—i.e., straightlining of responses—were minimal in the data, as Hair, Risher [63] suggest.

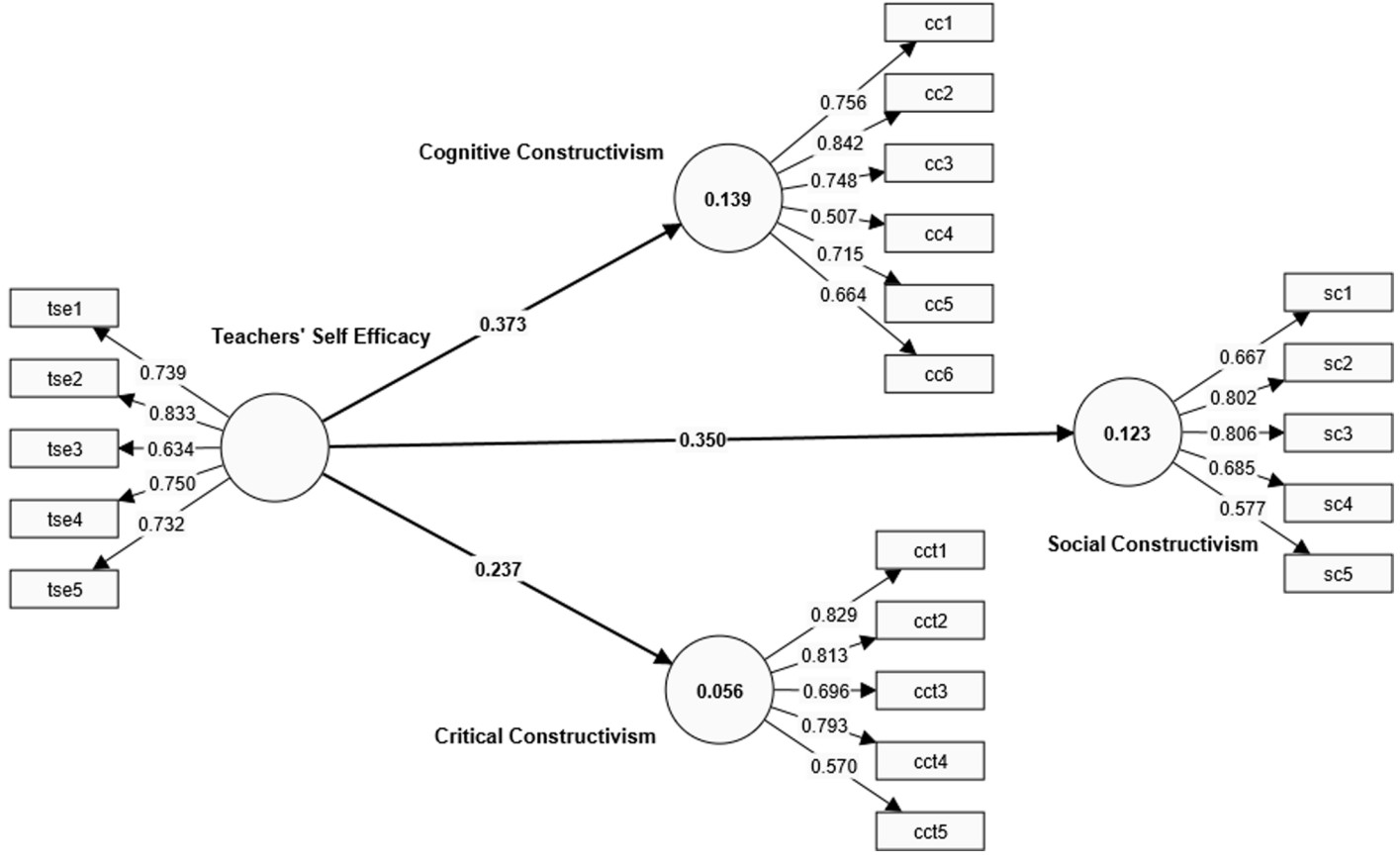

**Fig 2. PLS-SEM algorithm.**

The third consideration in the assessment of the measurement model in this study was the convergent validity analysis using the AVE. The AVE was used to ascertain the extent to which the construct converges to explain the variances in its latent variables. In other words, it was used to make a statistical claim that the construct indicators converge on a common underlying factor [65]. Practically, an AVE value of 0.5 or more is considered a good indication of convergent validity. In line with the recommended threshold, the constructs in this model are above the statistically acceptable AVE values. This shows, therefore, that the indicators measured a common construct.

**4.2.2. Discriminant validity assessment.** Discriminant validity analysis was also conducted to measure the quality of the proposed model. This validity analysis is concerned with ascertaining how the various constructs in the model are theoretically distinct from each other [66,67]. Establishing discriminant validity in PLS-SEM is usually based on the Fornell-Larcker criterion and the heterotrait-monotrait criterion. The current model's discriminant validity based on the three aforementioned criteria is analyzed in the subsequent paragraphs.

Based on the Fornell-Larcker criterion, discriminant validity of the constructs is established when the square root of the AVE of each construct is higher than the interfactor correlations, as suggested by Fornell and Larcker [68]. In line with this criterion, it could be concluded that discriminant validity has been established. Table 3 presents the results in this regard.

Per the statistics presented in Table 3, the square root of the AVEs of all the constructs is higher than their correlations with other factors. This indicated, therefore, that the constructs

**Table 3. Fornell-Larcker criterion.**

|       | CC    | CCT   | SC    | TSE   |
|-------|-------|-------|-------|-------|
| CC    | **0.713** |       |       |       |
| CCT   | 0.258 | **0.747** |       |       |
| SC    | 0.276 | 0.25  | **0.713** |       |
| TSE   | 0.373 | 0.237 | 0.35  | **0.740** |

**Table 4. HTMT ratio.**

|       | CC    | CCT   | SC    | TSE |
|-------|-------|-------|-------|-----|
| CC    |       |       |       |     |
| CCT   | 0.284 |       |       |     |
| SC    | 0.311 | 0.276 |       |     |
| TSE   | 0.367 | 0.220 | 0.422 |     |

are very distinct from each other. It also indicates that the items designed to measure the constructs do measure their respective underlying constructs but not any other constructs. The HTMT ratio is also another means of establishing discriminant validity in PLS-SEM. Discriminant validity using the HTMT is established when the average of item correlations across constructs is significantly lower than the average of correlations of items measuring the same constructs. Discriminant validity is met when none of the HTMT correlations is higher than the 0.90 threshold [63,69]. Table 4 presents the HTMT of the current model.

As can be seen, all the HTMT values are below the recommended threshold. Thus, the HTMT criterion has also provided enough evidence of discriminant validity in the constructs of the proposed model.

**4.2.3. Multicollinearity assessment.** Assessing the variance inflation factor (VIF) is a prerequisite for the assessment of the structural link between the study variables. This condition must always be met to avoid biasing the regression results, as Hair, Risher [63] warn. VIF assessment checks for the existence of multicollinearity; i.e., it is used to check whether there is a high correlation between two or more predictor variables. The multicollinearity results of the current model are presented in Table 5.

The VIF values of the current study (see Table 5) therefore suggest that there are no multicollinearity issues in the data since the highest VIF value is 3.174, which is below the recommended threshold of 3.3, as suggested by Kock [70].

## 4.3. Structural model results

Following the assessment of the measurement model, the various hypotheses were tested using PLS-SEM bootstrapping method. The PLS-SEM bootstrapping aimed to ascertain how teachers' constructivist self-efficacy influenced their practice of the various constructivist teaching approaches. Using PLS-SEM bootstrapping, the researcher focused on the strength and direction of the relationship (**β**) between the exogenous and endogenous variables, the coefficient of determination ($R^2$), the effect size of the predicted relationship ($f^2$) and the predictive validity (**Q2**) of the model. Fig 3 and Table 6 present the results of the PLS-SEM bootstrapping.

Table 6 presents the results on how teachers' constructivist self-efficacy predicts their constructivist practices. The influence of teachers' sense of efficacy and their practice of cognitive constructivism ($\beta = 0.371$; $t = 5.521$; $p < .001$) was found to be statistically significant; research

Table 5. Multicollinearity result (inner model).

|      | TSE | CC    | CCT   | SC    |
|------|-----|-------|-------|-------|
| TSE  |     | 1.295 | 1.056 | 1.515 |
| CC   |     |       |       |       |
| CCT  |     |       |       |       |
| SC   |     |       |       |       |

hypothesis 1 was therefore supported. This indicates that an increase in teachers' sense of efficacy is likely to increase in the extent to which they use cognitive constructivist means of enhancing students' learning of indigenous languages. The effect size for this prediction (f2 = 0.162) indicates that the magnitude of the impact of teachers' efficacy on their practice of cognitive constructivism is moderate. As suggested by Cohen [71] an $f^2$ of 0.02, 0.15 or greater, and 0.35 or greater represent small, moderate, and large effect sizes, respectively. The R2 value of 0.139 is an indication that, out of a host of many factors that may influence teachers' practice of cognitive constructivism, only efficacy accounts for 13.9% of the variation in this endogenous variable.

As with cognitive constructivism, the test also confirmed that the teachers' self-efficacy statistically influenced their practice of social constructivism (β = 0.352; t = 4.287; p < .001); hence, research hypothesis 2 was also supported. This also indicates that teachers' sense

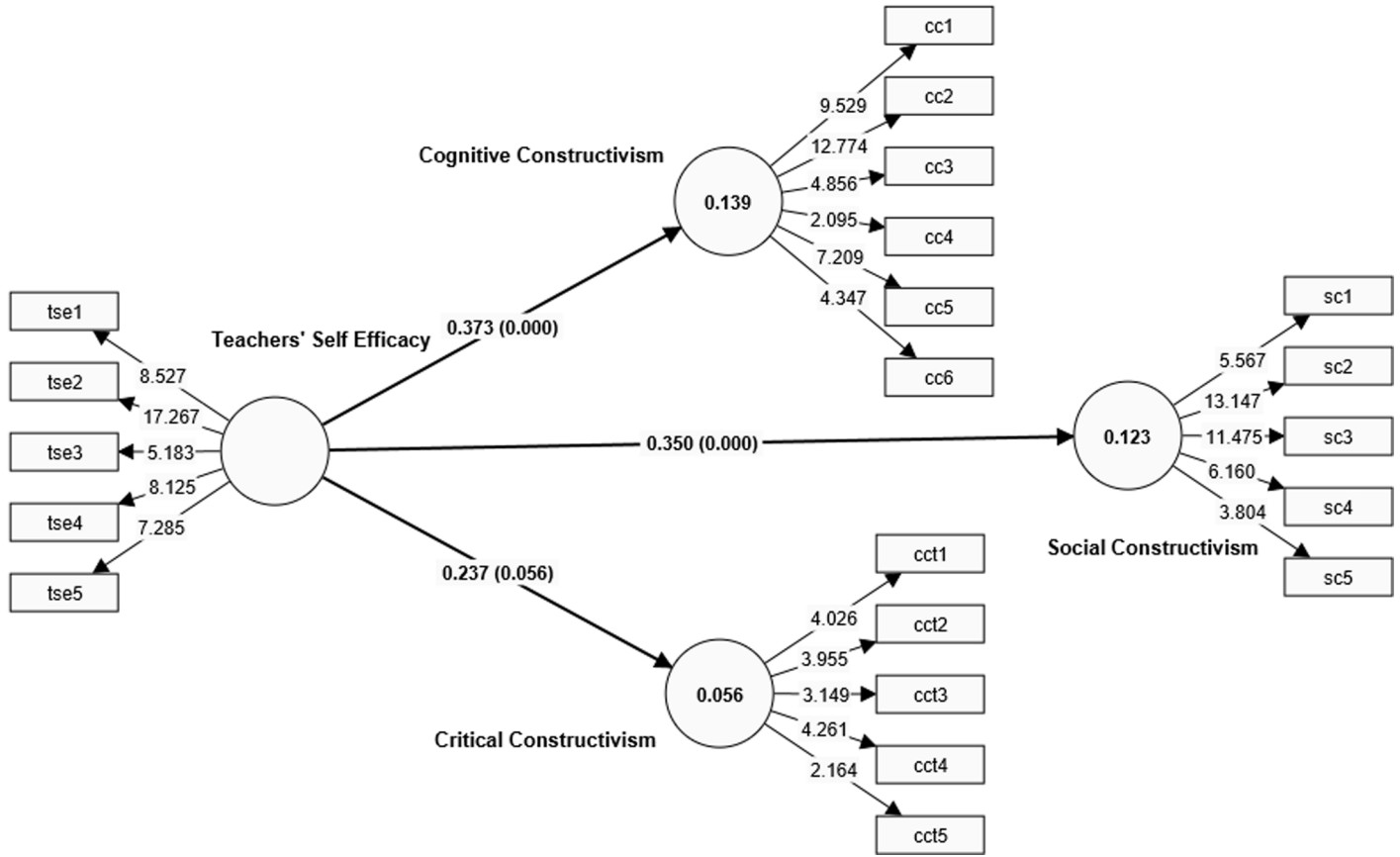

Fig 3. PLS-SEM bootstrapping results.

**Table 6. Bootstrapping results.**

| H | Path | β | SD | t | p | f2 | R2 | Q² | Decision |
|---|---|---|---|---|---|---|---|---|---|
| H1 | TSE -> CC | 0.373 | 0.066 | 5.637 | <.001 | 0.162 | 0.139 | 0.073 | Supported |
| H2 | TSE -> SC | 0.350 | 0.082 | 4.270 | <.001 | 0.140 | 0.123 | 0.028 | Supported |
| H3 | TSE -> CCT | 0.237 | 0.124 | 1.911 | 0.056 | 0.060 | 0.056 | 0.027 | Not Supported |

of efficacy tends to have an impact on the extent to which they create a language learning environment conducive to the enhancement of students' construction of knowledge through socially interactive means. The effect size ($f^2$) of 0.140 shows that teachers' sense of efficacy has a moderate influence on their enactment of social constructivism. The $R^2$ value indicates that efficacy accounts for 12.3% of the variation in social constructivism.

Unlike the first two forms of constructivism, the model's results indicate that teachers' self-efficacy does not statistically influence their practice of critical constructivism (β = 0.237; t = 1.199; p = 0.056); consequently, H3 is rejected. This statistical revelation suggests that irrespective of teachers' level of confidence as professional teachers, they are less likely to enact the principles of critical constructivism to enhance students' learning of the Ghanaian language and culture. Put succinctly, teachers' sense of confidence does not seem to compel them to give learners voice of authority in the classroom.

Finally, the predictive validity of the model was also assessed. According to Hair et al. (2019), predictive relevance is established in a model when the $Q^2$ exceeds zero. Precisely, a predictive relevance of 0, approximately 0.25, and 0.50 is an indication of small, medium, and large predictive relevance, respectively. In line with this, the $Q^2$ values of 0.073 for CC, 0.028 for SC, and 0.027 for CCT show that predictive relevance was established for all of the endogenous variables.

## 4.4. Qualitative findings

In line with Creswell and Clark [72], it is very crucial to provide a comprehensive explanation of quantitative results using qualitative data. In the current study, the primary goal of the follow-up qualitative inquiry was to understand why teachers' efficacy was statistically significant with all the variants of constructivist practices except for critical constructivism. It could be inferred from the qualitative results that socio-cultural concerns probably affected teachers' efficacy and the enactment of critical constructivism. Two major sociocultural issues that probably explain the lack of statistical relationship between teachers' sense of efficacy and the enactment of critical construction are the *adultism* and *resistance to change*.

**4.4.1. Adultism.** Teachers' views from the qualitative inquiry suggest that instructional principles that grant learners the voice to critique teachers' pedagogical plans and, most importantly, critique authoritative instruction, norms, and knowledge conflict with the cultural means of educating a child in the Ghanaian context. This is evident on the excerpt below:

> The Ghanaian culture is quite different. We all know that when an adult is at fault, you can not explicitly tell an adult that s/he is at fault. However, there are polite ways of telling the adult to admit his/her fault.

This and similar other submissions offered by the respondents show that teachers are less inclined, or better yet, less confident, about executing an instructional approach that breaches the existing norm of Ghanaian society. The implication of this is that, even if teachers have the

confidence to execute critical constructivism in language teaching, they would like to do so by applying the concept of politeness to make sure the cultural principles of raising the Ghanaian child are adhered to. The foregoing suggests the presence of adultism in Ghanaian education. As evident in the excerpt above, most language teachers who share similar views tend to believe that the principles of critical constructivism fight the inherited Ghanaian culture. This is quite a sad revelation and could perpetuate suppression of children's rights because it appears teachers are aware that in some instances their deeds and educational practices could be wrong and threat to students' academic wellbeing; nonetheless, by the virtue of being children they may not be given the opportunity to voice out their concerns because the culture frowns upon the critical voice of children.

**4.4.2. Resistance to change.** Teachers view critical constructivism as a weird instructional approach. When asked whether teachers would ever create an instructional environment where the critical voice of learners would be heard and given the necessary consideration in the language classroom, a teacher clearly stated:

> "That's not how it is done. "Throughout our educational journey, I have never seen a teacher ask learners to make decisions on which instructional technique to employ."

The view of this teacher shows clearly that there is a limited likelihood that they would ever shift from the authoritative educational paradigm to a more libertarian way of bringing up children in Ghanaian education. This view implies that irrespective of the magnitude of confidence teachers may develop, they are less likely to accept and implement any form of education that grants the learner the opportunity to be critical and, most importantly, an active participant in instructional decision-making in the classroom.

## 5. Discussions

This study modelled the relationship between teachers' perceived efficacy and their use of constructivist approaches in indigenous language teaching. Precisely, we modelled how teachers' sense of efficacy predicts their practice of cognitive, social, and critical constructivism. The findings from the statistical modelling are not counterintuitive because they appear to reflect the realities as far as education in Ghana is concerned. From the model's results, it is safe to assume that a teacher's sense of efficacy predicts what we consider 'explicit constructivist practices', which include social and cognitive constructivism. We consider these forms 'explicit constructivist practices' because their core principles are explicitly outlined in the school curriculum being implemented by the teachers. Thus, teachers are conscious of these approaches and are likely to implement them as expected. This likely explains why their sense of confidence significantly predicted these practices. This revelation aligns with that of Nyamekye, Zengulaaru [20], who found that teachers believed that their instructional practices aligned well with the social and cognitive constructivist way of improving learning at the basic level of education in Ghana.

Nonetheless, teachers' sense of efficacy did not significantly predict the practice of critical constructivism, which we consider a 'peripheral constructivist practice'. This form of constructivism was considered peripheral in the context of this study because it deviates from the mere idea of developing children's cognitive abilities through a form of education that grants learners the democracy to explore knowledge, question suppressive social structures, and the liberty to question the validity and reliability of knowledge irrespective of its source. The insignificant statistical association between teachers' efficacy and critical constructivism, along with the insights from the qualitative inquiry, provides further evidence that strengthens our belief in existing scholarly assertions that African teachers are more likely to

reject instructional philosophies that conflict with the cultured way of bringing up children [22,73,74].

Ideally, the culture of Ghana and most other African countries expects children to grow submissively [75]. Since children are usually regarded as developing human beings, their views are likely to be considered immature by adults or more experienced ones [23,76]. Such cultured thoughts are probably ingrained in Ghanaian teachers and are subconsciously manifesting in the instructional approaches as well as how teachers treat their learners. In the Taiwanese educational context, Aldridge, Fraser [77] discovered similar issues. They concluded in their study that critical voice as an aspect of constructivist practices was less observed in the lessons due to the demand for respect from children. Teachers were unwilling to enact such a constructivist principle because they were very concerned about their authoritative status as teachers. This and other factors, not included in the model, could probably account for the non-significant association between teachers' efficacy and their enactment of critical voice as an aspect of constructivism. It follows, therefore, that no matter the degree of confidence basic school teachers develops in ensuring constructivist learning, there is little chance that they will give students an authoritative voice in the classroom setting.

It is noteworthy, however, that teachers' unwillingness to embrace the critical aspect of constructivism seems quite detrimental to achieving the core goals of the standards-based curriculum, i.e., producing literate, effective problem solvers and, most importantly, producing individuals who possess the ability to think critically [12]. We argue in this regard because the possibility of developing children to become critical thinkers and problem solvers, for instance, would be very challenging in an educational system where children's desire to explore knowledge, question authority, and partake actively in decision-making is interpreted as a deviation from the societal norm and thus frowned upon.

## 6. Managerial implication of the study

Given the above discourse, we offer some suggestions for policymakers, including the GES, NaCCA, and MoE, to take into consideration. We suggest that for an effective realization of the stated goals of the standards-based curriculum, there is a need for culturally sensitive training for teachers to understand that this is, perhaps, the time for them to surrender their traditional thoughts about how children should be nurtured.

Particularly, the NaCCA, in collaboration with the Ghana Tertiary Commission (GTEC), should consider revising all teacher professional development programmes in various universities and colleges of education to align with current innovations in the basic school curriculum. A premium must be placed on the integration of professional courses that recognize the importance of constructivist principles, including critical pedagogy. This will enhance the production of competent teachers who would help in achieving the overarching goals of the SBC—i.e., the development of learners' autonomous learning, critical thinking and problem-solving skills.

## 7. Limitations and recommendations for further studies

The current study offered insight into how teachers' perceived sense of constructivist efficacy influenced their enactment of three constructivist practices: cognitive, social, and critical constructivism. Despite the valuable insight from the findings of this study, there is a need for further exploration because the current study relied largely on quantitative methods, characterised by self-reported data from teachers. It is therefore recommended that in-depth qualitative methods such as classroom observations be used to cross-validate teachers' actual constructivist practices. Moreover, the study relied solely on language teachers at the basic level of education in

Ghana. Given the relevance of this instructional philosophy in all fields of study at the basic level of education, there is a need for further research in other subject areas to deepen our understanding of how this instructional philosophy is enacted across all disciplines.

Most importantly, the lack of association between teachers' efficacy and their practice of critical constructivism raises critical questions that warrant further exploration. The majority of the study's respondents are early career teachers (with 1 – 5 years of teaching experience) who might have limited agency to execute these constructivist instruction practices for the sake of various systemic barriers. We therefore recommend further and in-depth exploration to understand how years of teaching experience might moderate the relationship between efficacy and teachers' critical constructivist practices.

## Author contributions

**Conceptualization:** Ernest Nyamekye.

**Data curation:** Ernest Nyamekye.

**Formal analysis:** Ernest Nyamekye.

**Methodology:** Ernest Nyamekye.

**Project administration:** Ernest Nyamekye.

**Supervision:** Seth Asare-Danso, Emmanuel Amo Ofori.

**Validation:** Seth Asare-Danso, Emmanuel Amo Ofori.

**Writing – original draft:** Ernest Nyamekye.

**Writing – review & editing:** Ernest Nyamekye.

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
