## [Decision Letter · Decision Letter 0]

4 Nov 2024

PONE-D-24-34525Investigating the Influence of teachers’ constructivist self-efficacy on their practice of Constructivism in Ghanaian language instructionPLOS ONE

Dear Dr. Nyamekye,

Thank you for submitting your manuscript to PLOS ONE. After careful consideration, we feel that it has merit but does not fully meet PLOS ONE’s publication criteria as it currently stands. Therefore, we invite you to submit a revised version of the manuscript that addresses the points raised during the review process.

Academic Editor:

Introduction: 1. Authors should clearly present the problem and define the scope of teacher efficacy in constructivist teaching. 2. They also need to strengthen the introduction with more recent data, evidence, or updated references is shared. 3. Revise or remove redundant content in the introduction (e.g., structure listing) is noted in both. Literature Review: 1. Authors should include more recent studies to keep the review current. 2. They should also detail the context of constructivist practices, particularly in specific regions like Ghana. 3. Considering focusing on studies linking teacher self-efficacy with constructivist methods is highlighted as important. Methods: 1. Authors should clearly justify choosing mixed-methods design and more detail on the research procedure is present in both reviews. 2. Try to include more detail on data analysis methods and how both qualitative and quantitative data are integrated and are consistent. Results: 1. Authors should provide additional participant data and detailed context on qualitative data collection and analysis. 2. The use of visual aids (tables or figures) to present findings is suggested. Discussions & Conclusions: 1. It would be effective if the authors split the “discussion and conclusion” section into two independent sections and include detailed limitations and implications are common. 2. The need for clearer contributions, comparisons with past research, and more updated references are noted. 3. There is an urgent need to discuss alternative explanations and justify the interpretation of results.

We look forward to receiving your revised manuscript.

Kind regards,

Ashraf Atta Mohamed Safein Salem

Academic Editor

PLOS ONE

Journal requirements: When submitting your revision, we need you to address these additional requirements. 1. Please ensure that your manuscript meets PLOS ONE's style requirements, including those for file naming. The PLOS ONE style templates can be found at https://journals.plos.org/plosone/s/file?id=wjVg/PLOSOne_formatting_sample_main_body.pdf and https://journals.plos.org/plosone/s/file?id=ba62/PLOSOne_formatting_sample_title_authors_affiliations.pdf

Additional Editor Comments:

Thank you to the authors for selecting a topic that I believe will capture the interest of many readers. However, there are several aspects of the manuscript that need to be refined and completed. Below are specific concerns that should be addressed:

Introduction:

1. Authors should clearly present the problem and define the scope of teacher efficacy in constructivist teaching.

2. They also need to strengthen the introduction with more recent data, evidence, or updated references is shared.

3. Revise or remove redundant content in the introduction (e.g., structure listing) is noted in both.

Literature Review:

1. Authors should include more recent studies to keep the review current.

2. They should also detail the context of constructivist practices, particularly in specific regions like Ghana.

3. Considering focusing on studies linking teacher self-efficacy with constructivist methods is highlighted as important.

Methods:

1. Authors should clearly justify choosing mixed-methods design and more detail on the research procedure is present in both reviews.

2. Try to include more detail on data analysis methods and how both qualitative and quantitative data are integrated and are consistent.

Results:

1. Authors should provide additional participant data and detailed context on qualitative data collection and analysis.

2. The use of visual aids (tables or figures) to present findings is suggested.

Discussions & Conclusions:

1. It would be effective if the authors split the “discussion and conclusion” section into two independent sections and include detailed limitations and implications are common.

2. The need for clearer contributions, comparisons with past research, and more updated references are noted.

3. There is an urgent need to discuss alternative explanations and justify the interpretation of results.

Reviewers' comments:

Reviewer's Responses to Questions

**Comments to the Author**

1. Is the manuscript technically sound, and do the data support the conclusions?

Reviewer #1: Partly

Reviewer #2: Partly

2. Has the statistical analysis been performed appropriately and rigorously? 

Reviewer #1: Yes

Reviewer #2: Yes

3. Have the authors made all data underlying the findings in their manuscript fully available?

Reviewer #1: Yes

Reviewer #2: Yes

4. Is the manuscript presented in an intelligible fashion and written in standard English?

Reviewer #1: Yes

Reviewer #2: Yes

5. Review Comments to the Author

Reviewer #1: Recommendation: Major revision

Detailed Comments:

1. Introduction: 1) The manuscript presents a clear problem statement, emphasizing the resistance of teachers to educational change and the need to explore teachers’ sense of efficacy in the context of constructivist teaching. However, more specific data or updated references that quantify the extent of teacher resistance or highlight the impact of teacher-centered learning on student outcomes should be provided. 2) While the introduction hints at the study’s practical implications, it should more explicitly address how this research advances the theoretical understanding of constructivist teaching and teacher self-efficacy. What new insights does it offer? Such as theoretical advancements, methodological innovations, or new empirical evidence. 3) The last paragraph of the introduction, which outlines the structure of the paper, is somewhat redundant and less informative. Consider deleting the section that merely lists the content of the forthcoming sections. Instead, conclude the introduction with a compelling summary that emphasizes how the study’s design and methodology are specifically tailored to address the research questions.

2. Literature Review: 1) Section 2.1: While the overview is thorough, consider integrating more recent studies to ensure the review is current. Additionally, discuss how these constructs of constructivism have been specifically applied or adapted in the Ghanaian educational context or similar settings. 2) Section 2.2: Please include studies that have specifically examined the relationship between self-efficacy and the implementation of constructivist teaching methods. This will provide a more direct link to the research focus. Additionally, provide a more detailed classification and elaboration of previous studies, including the measurement methods used and the background of the participants (e.g., college teachers, middle school teachers). This detail will help to identify and articulate the gaps in the existing literature more effectively. 3) Section 2.3: Please end with a brief summary of how the literature review informs the research questions and the study’s hypotheses.

3. Methodology: 1) Section 3.1: Lack a rationale for choosing this particular mixed-methods design over other possible designs. Discuss how this approach specifically addresses the research questions or hypotheses. Besides, the research procedure mentioned (Thus, in the context of ……) is too general, please add more details here or put the “research procedure” as an independent section. 2) Consider adding a section on data analysis methods, detailing how the quantitative and qualitative data will be analyzed separately and then integrated in the overall interpretation of results.

4. Results: 1) Provide more data about the participants, for example the mean value, SD, etc. 2) Provide more context on how the qualitative data were collected and analyzed. Detail the methods used for data collection (e.g., interviews, focus groups) and the data analysis process. Besides, consider using visual aids, such as tables or figures, to summarize key themes or findings from the qualitative data.

5. Discussions & Conclusions: 1) Structurally, please separate “Discussion and Conclusion” into two independent parts, in which discussion should include both limitations (Section 6) and implications (Section 7). 2) Expand on how this study contributes to or diverges from previous research, highlighting its unique contributions. Again, the references used in this part is too old to clarify your novel contributions to this field.

Reviewer #2: My thanks to the editor and authors for the opportunity to review this manuscript, which reports on the relationship between teachers self-efficacy regarding teaching using constructivism principles and the actual practice in the classroom in the Ghanaian context. This should be of interest to readers concerned with how teachers respond to national policy changes regarding pedagogy, and the qualitative component offers insight for relative success in such efforts.

There are several areas that I believe could be elaborated on to make the manuscript stronger.

1. The relationship between self-efficacy and instructional strategies in general is demonstrated in the literature review, but could the authors elaborate on the relationship between SE and respective forms of constructivism? Are there empirical studies operationalizing constructivist teaching in its various forms and show a prior relationship? The literature presents how SE is tied to teachers adopting new technologies and other practices, but there could be a stronger link made to SE and constructivism.

2. The authors cite a source indicating 6-12 participants as a standard for saturation in qualitative research, yet themselves collect data from 61. There is no standard for saturation in qualitative work, as satisfying that condition depends on the nature of the question, data type, and analysis. Can the author(s) clarify the decision-making process for including 61 participants and explicate how the specific qualitative data and analysis approach led to satisfaction that major insights had been exhausted?

3. The authors adapt two existing scales for use. Can the authors elaborate on the adaptation process and include sample items from the instrument? It is unclear from the manuscript how much change was made to the original instruments and what possible impact that might have on construct validity. This is particularly important for the CLES, as the authors only use 3 of 5 original dimensions and use those as proxies for the 3 types of constructivism. Readers may benefit from understanding what types of questions relate to respective forms of constructivism.

4. Can the authors elaborate on the semi-structured interview format and types of questions that were used to operationalize the various forms of constructivist teaching explored here? The nature of the qualitative data overall requires further explication, as well as the analytical process adopted.

5. In section 4.2, citations are needed for the adopted PLS-SEM analytical method used in the study.

6. The qualitative results reporting could also be more nuanced, as it is unclear how prevalent or salient particular insights are across the sample here and their representativeness for the study context overall.

7. Could the authors discuss alternative explanations or adjust the strength of the claim that the qualitative findings explain the unexpected results in the quantiative component? There might be other explanations (e.g., sample size, sample selection) or other factors influencing results. For example, most of the respondents indicated that hey had 1-5 years of teaching experience. The authors might elaborate on the relative agency of young teachers in this school system and consider if the minimal evidence for implementation is not a partial function of limited agency. Overall, the conclusions and implications as stated here are not sufficiently supported by the data and study limitations should be addressed.

8. There are a few formatting issues with citations, as well as a Creswell out of alphabetical order at the bottom.

9. The discussion adopts a first person singular voice briefly before entering first person plural. I suggest adopting a consistent voice throughout.

6. PLOS authors have the option to publish the peer review history of their article (what does this mean? ). If published, this will include your full peer review and any attached files.

**Do you want your identity to be public for this peer review?** For information about this choice, including consent withdrawal, please see our Privacy Policy .

Reviewer #1: No

Reviewer #2: No

---

## [Author Response · Author response to Decision Letter 1]

6 Dec 2024

Point-to-Point Response to Reviewers

Reviewer 1.

1. Introduction:

1) The manuscript presents a clear problem statement, emphasizing the resistance of teachers to educational change and the need to explore teachers’ sense of efficacy in the context of constructivist teaching. However, more specific data or updated references that quantify the extent of teacher resistance or highlight the impact of teacher-centered learning on student outcomes should be provided.

Response: this issue has been duly addressed. The last two paragraphs of the introduction review current studies in Ghana that provide empirical evidence that explains teachers' resistance as well as the factors that account for their unwillingness to align their instructional practices with the constructivist pedagogy (see pages 3 to 5)

2) While the introduction hints at the study’s practical implications, it should more explicitly address how this research advances the theoretical understanding of constructivist teaching and teacher self-efficacy. What new insights does it offer? Such as theoretical advancements, methodological innovations, or new empirical evidence.

Response: This issue has also been duly addressed. At the end of the introduction, we have provided a theoretical contribution to the study which makes it quite novel and distinct from the existing studies on constructivism

3) The last paragraph of the introduction, which outlines the structure of the paper, is somewhat redundant and less informative. Consider deleting the section that merely lists the content of the forthcoming sections. Instead, conclude the introduction with a compelling summary that emphasizes how the study’s design and methodology are specifically tailored to address the research questions.

Response: This suggestion has been taken into consideration. The last paragraph has been deleted and at the end of the last paragraph a concise summary of the data analytic approach employed to answer the research objective has been provided

2. Literature Review:

1) Section 2.1: While the overview is thorough, consider integrating more recent studies to ensure the review is current. Additionally, discuss how these constructs of constructivism have been specifically applied or adapted in the Ghanaian educational context or similar settings.

Response: The constructive suggestion has been taken into consideration. The last paragraph of the section shows how the various forms of constructivism have been adopted in the Ghanaian education system. We have cited the standards-based curriculum which advocates for the practices of constructivism.

Since the section primarily focused on the conceptual review with decided to review current studies on the practice of constructivism under the introduction section (see pages 4 to 5

2) Section 2.2: Please include studies that have specifically examined the relationship between self-efficacy and the implementation of constructivist teaching methods. This will provide a more direct link to the research focus. Additionally, provides a more detailed classification and elaboration of previous studies, including the measurement methods used and the background of the participants (e.g., college teachers, middle school teachers). This detail will help to identify and articulate the gaps in the existing literature more effectively.

Response: We appreciate the reviewer’s suggestion on this. We have duly resolved the issues raised. On pages 10 and 11, we have provided empirical literature that particularly looks at studies on the influence of efficacy on constructivist teaching and learning

3) Section 2.3: Please end with a brief summary of how the literature review informs the research questions and the study’s hypotheses.

Response: The suggestion has been taking into consideration and duly implemented

3. Methodology:

1) Section 3.1: Lack a rationale for choosing this particular mixed-methods design over other possible designs. Discuss how this approach specifically addresses the research questions or hypotheses. Besides, the research procedure mentioned (Thus, in the context of ……) is too general, please add more details here or put the “research procedure” as an independent section.

Response: The issue pointed out has been taken into consideration. We have rewritten the entire paragraph to make the rationale and justification for using explanatory sequential mixed method a bit clearer to potential readers of the manuscript.

2) Consider adding a section on data analysis methods, detailing how the quantitative and qualitative data will be analyzed separately and then integrated in the overall interpretation of results.

Response: We thank the reviewer for point this significant gap in our methods sections. We have developed a subsection that comprehensively present data the processing and analysis technique followed in the current study (see pages 15 and 16)

4. Results:

1) Provide more data about the participants, for example the mean value, SD, etc.

Response: This seems like a tangible suggestion from the reviewer. However, the specific direction the reviewer gave is quite unclear. We do not know whether the reviewer refers to the mean age of the respondents or not. That notwithstanding, we believe the presenting participants’ demographic data in frequency and percentages is more easily comprehensible than in the form of means and standard deviation, especially to readers with little statistical knowledge. Thus, we prefer the presentation of participants demographic in the current statistical format

2) Provide more context on how the qualitative data were collected and analyzed. Detail the methods used for data collection (e.g., interviews, focus groups) and the data analysis process. Besides, consider using visual aids, such as tables or figures, to summarize key themes or findings from the qualitative data.

Response: We appreciate the reviewer’s contributions. However, we plead to oppose this particular suggestion because this issue has been resolved in the methods section. We pointed out that the qualitative data were elicited through interviews. From our point of view, detailing the data collection processes in the results section as the reviewer suggests could create some form of redundancy. Also, is quite difficult for us to present qualitative data in the form of visual aids, such as tables or figures. Besides, this is not a qualitative dominant study and there we only considered discussing a few quotes which explained the uncertainties in the quantitative results

5. Discussions & Conclusions:

1) Structurally, please separate “Discussion and Conclusion” into two independent parts, in which the discussion should include both limitations (Section 6) and implications (Section 7).

Response: We acknowledge the relevance of the reviewers’ comments. Nonetheless, the discussion, especially the last two paragraphs somewhat concludes the study. Thus, getting another section for conclusion would create redundancy which would eventually make reading the paper kind of boring. Moreover, the implication of the study section also sort of presents conclusions and recommendations for policy and practice. Per the structure and discussion in the discussion section and the implication section, developing a separate section for ‘conclusion’ would create redundancy.

2) Expand on how this study contributes to or diverges from previous research, highlighting its unique contributions. Again, the references used in this part is too old to clarify your novel contributions to this field.

Response: Thank you for the contribution. However, we believe the reviewers' comments are already in the discussion of the study. In the discussion of the results, we have discussed how our statistical contribution to knowledge already connects with qualitative discourse on the subject matter, citing several works by Nyamekye et al. (2023(Ndofirepi & Shumba, 2012; Nthontho, 2017; Nyamekye, Zengulaaru, Addae, Mutawakil, & Ntiakoh, 2024a)(Cross & Ndofirepi, 2015; Nyamekye, Mutawakil, et al., 2024). Moreover, these works are some of the few recently published works on the subject under investigation. Frankly, there seem to be limited studies on the subject that are more recent than the above, especially in the context of Africa

Review 2.

1. The relationship between self-efficacy and instructional strategies in general is demonstrated in the literature review, but could the authors elaborate on the relationship between SE and respective forms of constructivism? Are there empirical studies operationalizing constructivist teaching in its various forms and show a prior relationship? The literature presents how SE is tied to teachers adopting new technologies and other practices, but there could be a stronger link made to SE and constructivism.

Response: Thank you for the suggestion. Based on the reviewers' recommendations, we have reviewed a few more studies that specifically looked at the link between self-efficacy and constructivism (see pages 10 and 11)

2. The authors cite a source indicating 6-12 participants as a standard for saturation in qualitative research, yet themselves collect data from 61. There is no standard for saturation in qualitative work, as satisfying that condition depends on the nature of the question, data type, and analysis. Can the author(s) clarify the decision-making process for including 61 participants and explicate how the specific qualitative data and analysis approach led to satisfaction that major insights had been exhausted?

Response: Thank you for pointing this out. We have revised this mistake. We misquoted the exact sample for the qualitative data. Originally, only 15 teachers willingly participated in this study. The 61 figure was an anticipated number of interviews we while the paper was in the proposal stage. We hope that the reviewer understands this uncertainty.

3. The authors adapt two existing scales for use. Can the authors elaborate on the adaptation process and include sample items from the instrument? It is unclear from the manuscript how much change was made to the original instruments and what possible impact that might have on construct validity. This is particularly important for the CLES, as the authors only use 3 of 5 original dimensions and use those as proxies for the 3 types of constructivism. Readers may benefit from understanding what types of questions relate to respective forms of constructivism.

Response: The comments and suggestions have been duly addressed. As could be seen on pages 14 and 15, we have outlined the adaption processes and given samples of the items that were adapted from the CLES to measure the constructs in the current study. We have also provided a justification for adapting the scales, citing the explanations offered by Taylor et al., 1997

4. Can the authors elaborate on the semi-structured interview format and types of questions that were used to operationalize the various forms of constructivist teaching explored here? The nature of the qualitative data overall requires further explication, as well as the analytical process adopted.

Response: In page 15, the issue pointed out by the reviewer has been addressed. We have explicated the issues that were explored to gather the qualitative data.

5. In section 4.2, citations are needed for the adopted PLS-SEM analytical method used in the study.

Response: In the section, several authorities in the context of PLS-SEM have been cited (Hair, Risher, Sarstedt, & Ringle, 2019; Jöreskog, 1971) to explain the rationale behind the use of specific statistical techniques

6. The qualitative results reporting could also be more nuanced, as it is unclear how prevalent or salient particular insights are across the sample here and their representativeness for the study context overall.

Response: We thank the reviewer for the suggestion. Our position on this issue is that the qualitative results was not necessary used for purpose of generalization. However, the main was to gain further insight to supplement and explain the statistical results. Hence, the qualitative findings were reported with caution. As evident in the presentation of the qualitative finding, we claim that “ It could be inferred from the qualitative results that socio-cultural concerns probably affected teachers’ efficacy and the enactment of critical constructivism.” (p. 27) This suggests that we did not mean to emphatically and objectively state that teachers’ qualitative data provides an objective and absolute view of the lack of significant association between the efficacy and constructivist practices

7. Could the authors discuss alternative explanations or adjust the strength of the claim that the qualitative findings explain the unexpected results in the quantitative component? There might be other explanations (e.g., sample size, sample selection) or other factors influencing results. For example, most of the respondents indicated that they had 1-5 years of teaching experience. The authors might elaborate on the relative agency of young teachers in this school system and consider if the minimal evidence for implementation is not a partial function of the limited agency. Overall, the conclusions and implications as stated here are not sufficiently supported by the data and study limitations should be addressed.

Response: We thank the reviewer for this constructive contribution. Despite the relevance of this suggestion, our position is that the teaching experience could also offer plausible explanation to lack of statistical association between efficacy and the practice of constructivism among language teachers. However, since we do not want to make such theoretical projections without data in the current study, we have considered addressing this as part of the study limitation and a direction for further exploration

8. There are a few formatting issues with citations, as well as a Creswell out of alphabetical order at the bottom.

Response: The citation format has been entirely revised, hence resolving the above issue

9. The discussion adopts a first-person singular voice briefly before entering first-person plural. I suggest adopting a consistent voice throughout.

Response: Thank you for the comment, we have considered the suggest and have made the necessary change

---

## [Decision Letter · Decision Letter 1]

17 Feb 2025

Investigating the Influence of teachers’ constructivist self-efficacy on their practice of Constructivism in Ghanaian language instruction

PONE-D-24-34525R1

Dear Dr. Ernest Nyamekye,

We’re pleased to inform you that your manuscript has been judged scientifically suitable for publication and will be formally accepted for publication once it meets all outstanding technical requirements.

Kind regards,

Ashraf Atta Mohamed Safein Salem

Academic Editor

PLOS ONE

Additional Editor Comments (optional):

Reviewers' comments:

Reviewer's Responses to Questions

**Comments to the Author**

1. If the authors have adequately addressed your comments raised in a previous round of review and you feel that this manuscript is now acceptable for publication, you may indicate that here to bypass the “Comments to the Author” section, enter your conflict of interest statement in the “Confidential to Editor” section, and submit your "Accept" recommendation.

Reviewer #3: (No Response)

2. Is the manuscript technically sound, and do the data support the conclusions?

Reviewer #3: Yes

3. Has the statistical analysis been performed appropriately and rigorously? 

Reviewer #3: Yes

4. Have the authors made all data underlying the findings in their manuscript fully available?

Reviewer #3: Yes

5. Is the manuscript presented in an intelligible fashion and written in standard English?

Reviewer #3: Yes

6. Review Comments to the Author

Reviewer #3: Manuscript Number: PONE-D-24-34525R1

Manuscript Title: Investigating the Influence of teachers’ constructivist self-efficacy on their practice of Constructivism in Ghanaian language instruction

The study is beneficial and timely for educational discourse within the Ghanaian context. It seeks to improve the policy and practice in Ghana. However, certain aspects of the paper need to be tightened and reviewed to improve the quality. Reviewer 1 and 2 raised concerns that needs to be addressed.

Methodology

The data analysis of the qualitative data needs further elaboration. The interpreted meaning, synthesis and summary of the analysis for presentation are missing in the sub-section.

The reviewer 2 in number 4 raised an issue on the elaboration on the semi-structured interview format and types of questions that were used to operationalised the various forms of constructivist teaching explored in the study. This has not been addressed by the authors in the study.

Results

The point 2 raised by reviewer 1 is very significant to the quality of the manuscript and the authors are supposed to address that.

The reviewer 2 raised an issue of detailing the qualitative results of the study. This has not been addressed and needs to be given the necessary attention by the authors.

The point 7 raised by reviewer 2 needs to be addressed by the authors.

7. PLOS authors have the option to publish the peer review history of their article (what does this mean? ). If published, this will include your full peer review and any attached files.

**Do you want your identity to be public for this peer review?** For information about this choice, including consent withdrawal, please see our Privacy Policy .

Reviewer #3: **Yes: ** Dr Enock Swanzy-Impraim

---

## [Editor Report · Acceptance letter]

PONE-D-24-34525R1

PLOS ONE

Dear Dr. Nyamekye,

I'm pleased to inform you that your manuscript has been deemed suitable for publication in PLOS ONE. Congratulations! Your manuscript is now being handed over to our production team.

Kind regards,

on behalf of

Dr. Ashraf Atta Mohamed Safein Salem

Academic Editor

PLOS ONE